# Ocean-bottom seismometers reveal surge dynamics in Earth's longest-runout sediment flows
Pascal Kunath [1] ✉, Peter J. Talling [2,3], Dietrich Lange [1], Wu-Cheng Chi[4], Megan L. Baker [3], Morelia Urlaub [1,5] & Christian Berndt [1]

Turbidity currents carve Earth's deepest canyons, form Earth's largest sediment deposits, and break seabed telecommunications cables. Directly measuring turbidity currents is notoriously challenging due to their destructive impact on instruments within their path. This is especially the case for canyon-flushing flows that can travel >1000 km at >5 m/s, whose dynamics are poorly understood. We deployed ocean-bottom seismometers safely outside turbidity currents, and used emitted seismic signals to remotely monitor canyon-flushing events. By analyzing seismic power variations with distance and signal polarization, we distinguish signals generated by turbulence and sediment transport and document the evolving internal speed and structure of flows. Flow-fronts have dense near-bed layers comprising multiple surges with 5-to-30-minute durations, continuing for many hours. Fastest surges occur 30–60 minutes behind the flow-front, providing momentum that sustains flow-fronts for >1000 km. Our results highlight surging within dense near-bed layers as a key driver of turbidity currents' long-distance runout.

Turbidity currents are the longest-runout sediment-driven flows on Earth[1], playing a key role in shaping Earth's deepest and longest canyons and forming its largest sediment accumulations[2–4]. In a few hours, turbidity currents can transport more sediment mass to the deep sea than the global annual mass flux from all rivers combined[1,5,6]. These powerful flows can reach speeds of 20 m s$^{-1}$ and travel over 1000 kilometers[1]. They link rivers to the deep ocean and impact geology, biology, and climate on a global scale[7,8]. However, they also pose a major threat as they frequently break seabed telecommunication cable networks that carry over 95% of global intercontinental data traffic, underpinning many aspects of daily life such as the internet, financial markets, and cloud data storage[9].

Monitoring turbidity currents had long been considered impractical due to their destructive and unpredictable nature[10]. To date, direct measurements are mostly limited to slower (<2–5 m s$^{-1}$), short-runout flows (<50 km) in shallow waters (<2 km depth) at about 12 sites worldwide[1,11–16]. More powerful currents destroyed moored instruments[5,13], leading to the loss of equipment and data. Consequently, measurements of larger, less frequent turbidity currents—which carve submarine canyons, dominate sediment and carbon transport to the deep-ocean, and pose the greatest hazards to cables—are scarce. Previous measurements of such flows have come from cable breaks or destroyed moorings at just two sites[5,17], providing only estimates of flow front (transit) speeds and run-out distances, but offering limited insight into their internal structure.

Resolving the internal structure of canyon-flushing turbidity currents is crucial for predicting their dynamics, impact on seafloor infrastructure, and deposit architecture[18–24]. Powerful turbidity currents are thought to be driven by fast, dense near-bed layers[12,13,24], but it is uncertain whether these layers move continuously or surge dynamically, similar to terrestrial debris flows and snow avalanches[23,25,26]. This distinction matters because continuous and surging flows affect sediment suspension, bed friction, and impact forces on the ground differently—factors influencing erosion, flow velocity, runout distance, and hazard potential.

On land, remote seismic monitoring has revolutionized our understanding of major geohazards such as floods, debris flows, glacial lake outbursts, and avalanches, by detecting their ground motions via seismometers with millisecond precision across distances ranging from hundreds of meters to hundreds of kilometers[27–29]. These data have yielded key insights into how ground motion signals are generated at the source, transmitted through the environment, and ultimately recorded at seismic stations, thereby advancing process understanding, disaster response, and early warning systems[29]. In submarine settings, ocean-bottom seismometers (OBSs) and hydrophones have occasionally recorded seismic and acoustic

[1]GEOMAR Helmholtz Centre for Ocean Research, Kiel, Germany. [2]Department of Earth Sciences, Durham University, Durham, UK. [3]Department of Geography, Durham University; South Road, Durham, UK. [4]Institute of Earth Sciences, Academia Sinica, Taipei, Taiwan. [5]Kiel University, Christian-Albrechts-Platz 4, Kiel, Germany. ✉e-mail: pkunath@geomar.de

signals from submarine mass movements[30–33], but their use remains nascent and often limited to detecting occurrence and overall duration.

Here, we present the first detailed measurements of the internal structure, speed, and spatiotemporal evolution of dense-frontal cells in canyon-flushing turbidity currents, thereby going beyond previous measurements restricted to just their front-speed, runout-distance or total duration[5,33]. These results were obtained by analyzing seismic signals recorded by OBSs positioned safely outside the Congo Canyon and Channel off West Africa.

Our study has three objectives: first, to understand how turbidity currents generate seismic signals, testing the hypothesis that these signals arise from flow turbulence and sediment transport; second, to show how seismic data can track the location and velocity of these flows, revealing internal sediment pulses with varying speeds; and third, to underpin a new view for the structure and internal dynamics of canyon-flushing turbidity currents based on these findings.

## Results

### Turbidity currents in the Congo Canyon and Channel

The Congo Canyon begins within the estuary of the Congo River (Fig. 1), which ranks second in water discharge and fifth in particulate organic carbon export among the world's rivers[34]. The submarine canyon exhibits pronounced relief (up to 1200 m) along the continental shelf for the first ~150 km. It then transitions into a less-incised deep-sea channel (250–150 m deep) with depositional levees, which terminates 1100 km from the river mouth at a depositional lobe. From October 2019 to May 2020, OBSs were deployed along the canyon-channel system, on terraces or levees 0.5 to 3.0 km outside the canyon-channel axis, at locations OBS1 to OBS10 (Fig. 1). These OBS were complemented by moorings with Acoustic Doppler Current Profilers (ADCPs) deployed inside the canyon-channel-floor[5]. For slower turbidity currents (<2–3 m s$^{-1}$) that did not break the ADCP moorings[5], ADCP-derived velocities were compared to seismic data from adjacent OBS sites, providing benchmarks for the OBS data[33]. These comparisons revealed that the seismic signals originate from the faster-moving dense frontal zone, which outpaced the slower-moving dilute flow body[33].

The ADCP-moorings were subsequently broken by powerful turbidity currents, including a major canyon-flushing event on January 14–16, 2020. This event also broke a series of telecommunication cables, disrupting internet and data transfer across large parts of Africa during the COVID-19 pandemic. A second canyon-flushing, cable-breaking flow occurred on March 8, 2020. Transit speeds from cable breaks, ADCPs, and OBSs showed that the fronts of these turbidity currents traveled at 5–8 m s$^{-1}$ over 1100 km[5,33], making them the longest-runout sediment flows yet measured in action on Earth. They eroded ~2.68 km$^3$ of sediment, equivalent to 19–33% of the global sediment flux from all rivers to the ocean[5,35]. The terrestrial organic carbon transported by these two turbidity currents rivals the estimated amount buried globally in oceans each year[6]. Remarkably, these flows maintained near-steady speeds, and the duration of seismic signals changed

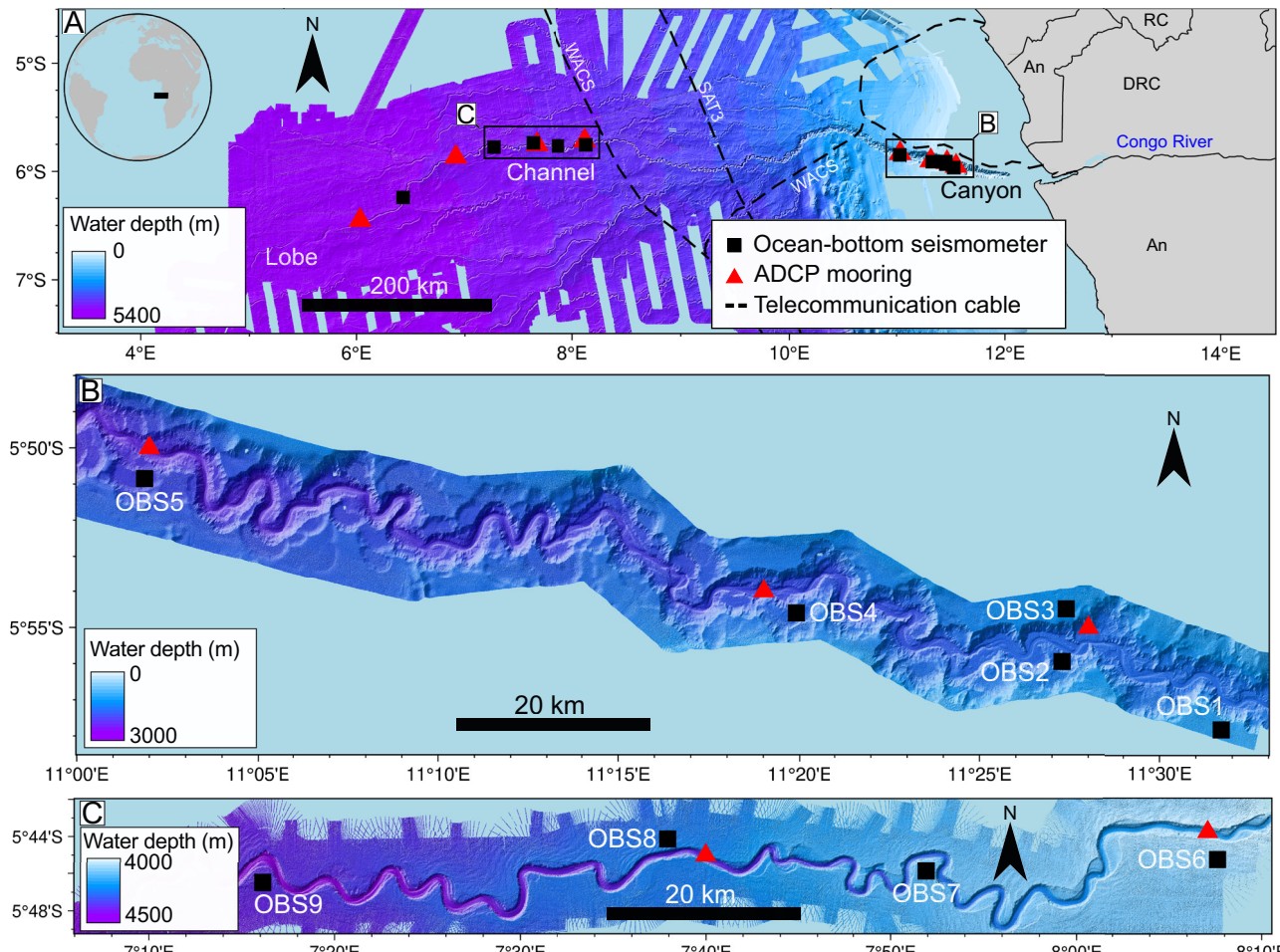

**Fig. 1 | Bathymetric map of the Congo Canyon-Channel system and instrument placement. A** Bathymetric map of the Congo Canyon-Channel system, located offshore West Africa (location in inset), showing the placement of instruments in the canyon (**B**) and channel (**C**) sub-arrays. Acoustic Doppler current profiler (ADCP) moorings (red triangles) were located in the channel axis, where they were eventually broken by powerful canyon-flushing turbidity currents (together with the South Atlantic 3 (SAT-3) and West Africa Cable System (WACS) cables), whilst the ocean-bottom seismometers (OBSs; black squares) were located on the canyon terraces and channel thalweg, out of harm's way. An, Angola; DRC, Democratic Republic of the Congo; RC, Republic of the Congo.

only moderately over these long distances, despite this substantial seabed erosion[5,33]. This challenges previous ignition theory that inferred pronounced seabed erosion would cause a turbidity current to become denser and faster, and lead to yet more erosion and further acceleration[19].

## The seismic footprint of turbidity currents

We identified turbidity currents by continuous ground motions, with dominant frequencies ranging between 1–7 Hz (Fig. 2). The ground motions display emergent waveforms, characterized by a gradual build-up to a peak amplitude later in the signal, followed by a subsequent gradual decay. Peak vertical ground motion amplitudes varied from $10^{-7}$ to $10^{-5}$ m s$^{-1}$, with durations ranging from 30 minutes to 14 hours. Hydrophones mounted on the OBSs failed to record signals from the turbidity currents[33].

We observed variations in ground motion waveforms and spectral signatures between different stations and events, despite generally consistent patterns. For example, the turbidity current recorded on March 8th at OBS4 exhibited a relatively smooth, emergent high-frequency ground motions with dominant amplitudes confined to the 1–3.5 Hz range (Fig. 2C). The ground motions reached its maximum amplitude and spectral frequency of

7 Hz within an hour, followed by slow tapering over the next four hours. In contrast, the same event recorded at OBS6 (Fig. 2D), located 500 km downslope, initially displayed high-frequency ground motions with weak amplitudes in the 1–3.5 Hz range, but within 15 to 30 minutes, it broadened to cover 1–5 Hz, with substantial amplitude variability. The signal's power surged, peaking for 15 minutes at a spectral frequency of 7 Hz, before narrowing to 5 Hz and gradually decaying.

## Spectral characteristics of turbidity current sediment transport

We first analyze how turbidity currents generate seismic signals. Seismic records were analyzed from the same flow for adjacent OBS at different distances from the canyon axis, such as OBS2 that is 800 meters from the canyon-axis, and OBS3 that is 1600 meters from the canyon-axis. This analysis assumes consistent signal sources and ground properties, making distance to the canyon the only variable. This approach allows us to distinguish between signal sources based on differences in their spectral signatures[29]. To interpret potential signal sources, we compare these signatures to those produced by turbulent flow[36] and fluvial bedload transport[37] models applied to the marine environment (see methods).

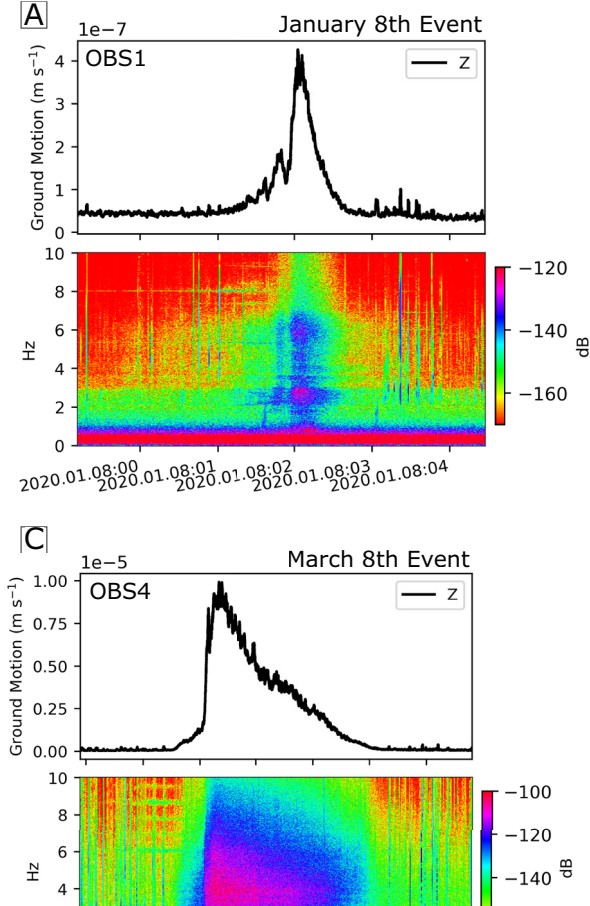

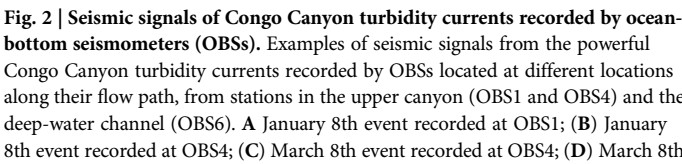

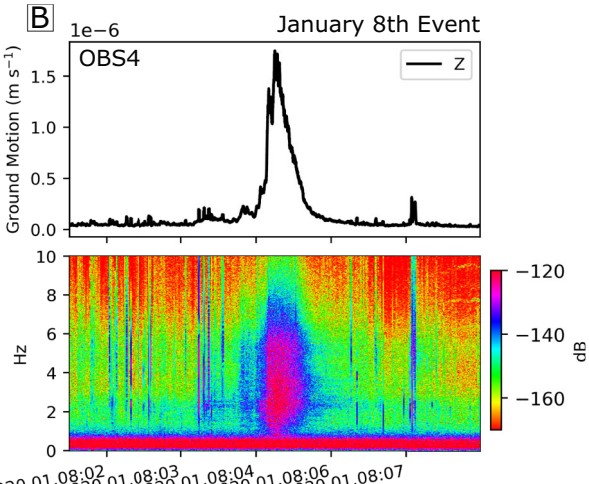

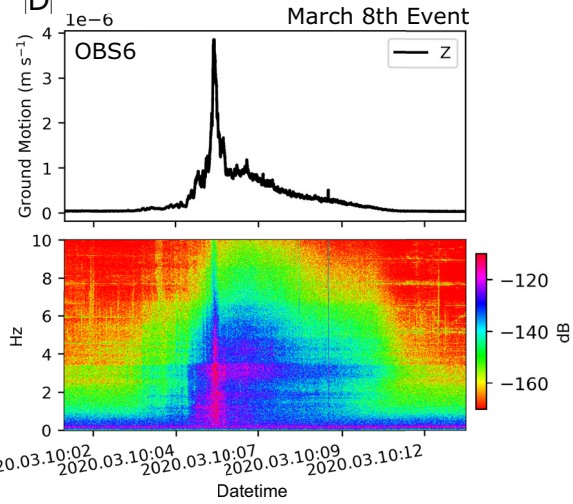

**Fig. 2 | Seismic signals of Congo Canyon turbidity currents recorded by ocean-bottom seismometers (OBSs).** Examples of seismic signals from the powerful Congo Canyon turbidity currents recorded by OBSs located at different locations along their flow path, from stations in the upper canyon (OBS1 and OBS4) and the deep-water channel (OBS6). **A** January 8th event recorded at OBS1; (**B**) January 8th event recorded at OBS4; (**C**) March 8th event recorded at OBS4; (**D**) March 8th event recorded at OBS6. The upper panel for each event displays the waveform envelope of high-frequency (>10 Hz) vertical ground motion recorded from approximately 2 hours before the turbidity current. The corresponding spectrogram is shown in the lower panel. The seismic signals of turbidity currents reveal a consistent pattern of an emergent phase, followed by a peak and a decay.

Our analysis reveals two distinct frequency bands (Fig. 3) from individual events at OBS2 and OBS3: a 1–3 Hz band likely associated with flow turbulence, and a 4–6 Hz likely linked to sediment transport. This attribution is based on insights from seismic studies of fluvial sediment transport[29,36–38], which suggest that stations closer to the channel are more sensitive to higher-frequency signals from sediment transport, with this sensitivity decreasing at greater distances. Conversely, signals from turbulent flow remain dominant at lower frequencies and persist over greater distances. At OBS2 (Fig.3B), the spectra show increased power in the 4–6 Hz range, which diminishes at the more distant OBS3 (Fig. 3A). However, the 1–3 Hz band remains prominent at both stations, supporting such an interpretation.

Furthermore, modeling predicts that if the signals were generated exclusively by a single process, such as turbulence or bedload transport, the spectral differences between OBS2 and OBS3 would exhibit a monotonous decrease in seismic power with increasing frequency due to signal attenuation (Fig. S1). However, the observed spectral differences between OBS2 and OBS3 show a non-monotonic behavior. There is a distinctive notch around 4 Hz marked by a drop in seismic power, flanked by higher energy levels (Fig. 3C), deviating from theoretical expectations. This notch pattern, also observed for the other canyon-flushing turbidity current (Fig. S2), likely arises from spectral overlap where turbulence and sediment transport dominate at different frequencies and distances, as suggested by the physics-based models (Fig. 3D, E).

### Detection of multiple surges

These field observations show that the flow front of canyon-flushing turbidity currents consists of multiple surges (Fig. 4). Each surge is recorded by a signal source moving down the canyon, causing the direction from the OBS to the source to progressively rotate from up-canyon to down-canyon as the surge passes the OBS location.

The signal sources are located by analyzing the polarization of Scholte waves emitted by turbidity currents (Fig. S3), which provides the back-azimuth (BAZ) direction (see Methods). The source location is then inferred where this BAZ direction intersects the canyon axis.

For instance, Fig. 4B illustrates the evolution of BAZ during the March 8th event. The event was initiated at 4:45 am, indicating the onset of the flow's emergent phase. Initially, the BAZ shifted from 40 degrees (northeast) to 340 degrees (northwest) over 20 minutes before returning to the northeast (Fig. 4A). This oscillating pattern repeated several times, with smaller azimuthal variations and more substantial shifts during certain moments, such as the waveform peak when the signal moved towards 40 degrees (northwest). During the event, the degree of polarization (DOP) increased from about 0.4 to 0.6 in the emergent phase and subsequently decreased to below 0.4 as the signal dissipated around 9:30 am. Individual surges last 5–20 minutes.

We interpret the systematic evolution in BAZ and DOP as evidence of multiple consecutive sediment surges within the turbidity current flow. These surges initially approach from the northeast, then move north, and subsequently shift westward. They are detectable only within a certain range of the seismometer; stronger surges extend farther, causing greater BAZ variability. This pattern was clearly observed during the March 8th event at OBS6, where the waveform peak was detected up to 5 km away. As the surge moves out of range, the BAZ gradually shifts upstream, suggesting a new surge is approaching and becoming dominant, while earlier surges fade downstream.

### Velocity transients in surges

By tracking the temporal changes in source locations along the canyon segments, we can calculate the speed of individual surges. For example, during the March 8th turbidity current at OBS6, surges had speeds ranging from 5 to 6.3 m s$^{-1}$, averaging 5.6 ± 0.8 m s$^{-1}$. This average surge speed closely matches the flow front's transit speed of 5.7 m s$^{-1}$ between OBS5 and OBS6, derived from arrival times. This alignment confirms the reliability of our surge speed measurements.

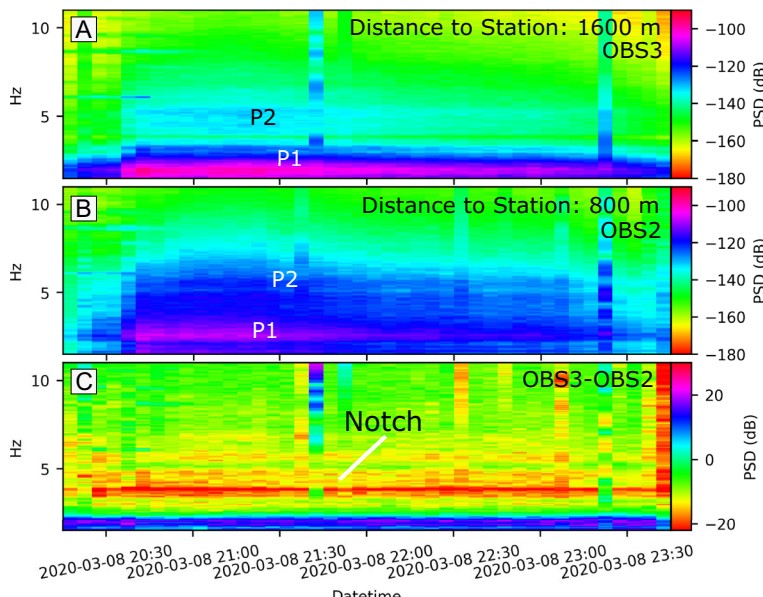 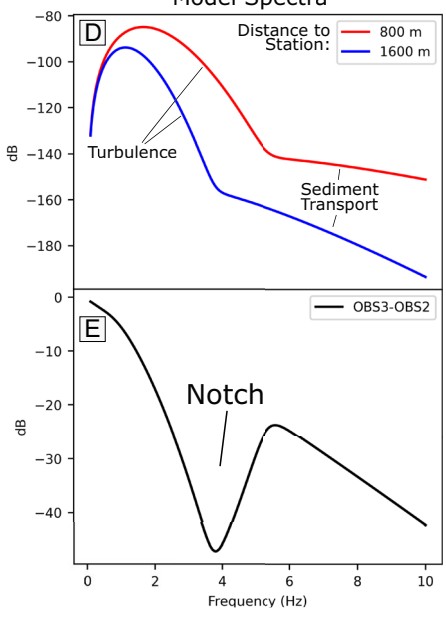

**Fig. 3 | Spectral characteristics of the March 8th turbidity current recorded at OBS2 and OBS3.** Spectrograms for the March 8th turbidity current recorded at (**A**) OBS3 and (**B**) OBS2, and their (**C**) difference. The OBS stations are located in close proximity, within <3 km distance from each other, but on opposite sides of the canyon (Fig. 1B for location). A comparison of the turbidity spectral signatures from OBS2 and OBS3 reveals two distinct phases: one at 1-3 Hz (P1) and another at 4-6 Hz (P2). At the larger distance, seismic power is concentrated in the 1–3 Hz phase, with the 4-6 Hz phase attenuated (**A**). In contrast, both phases show strong power at the shorter distance (**B**). The difference between the spectral signatures from both stations shows a non-monotonic behavior, with a distinct notch around 3−5 Hz (**D**, **E**). PSDs are generated by combining flow turbulence and sediment (bedload) transport based on fluvial models from Gimbert et al.[36] and Tsai et al.[37], using the source-to-station distances from OBS2 (red) and OBS3 (blue) as input, while assuming perfectly elastic sediment collisions (see methods).

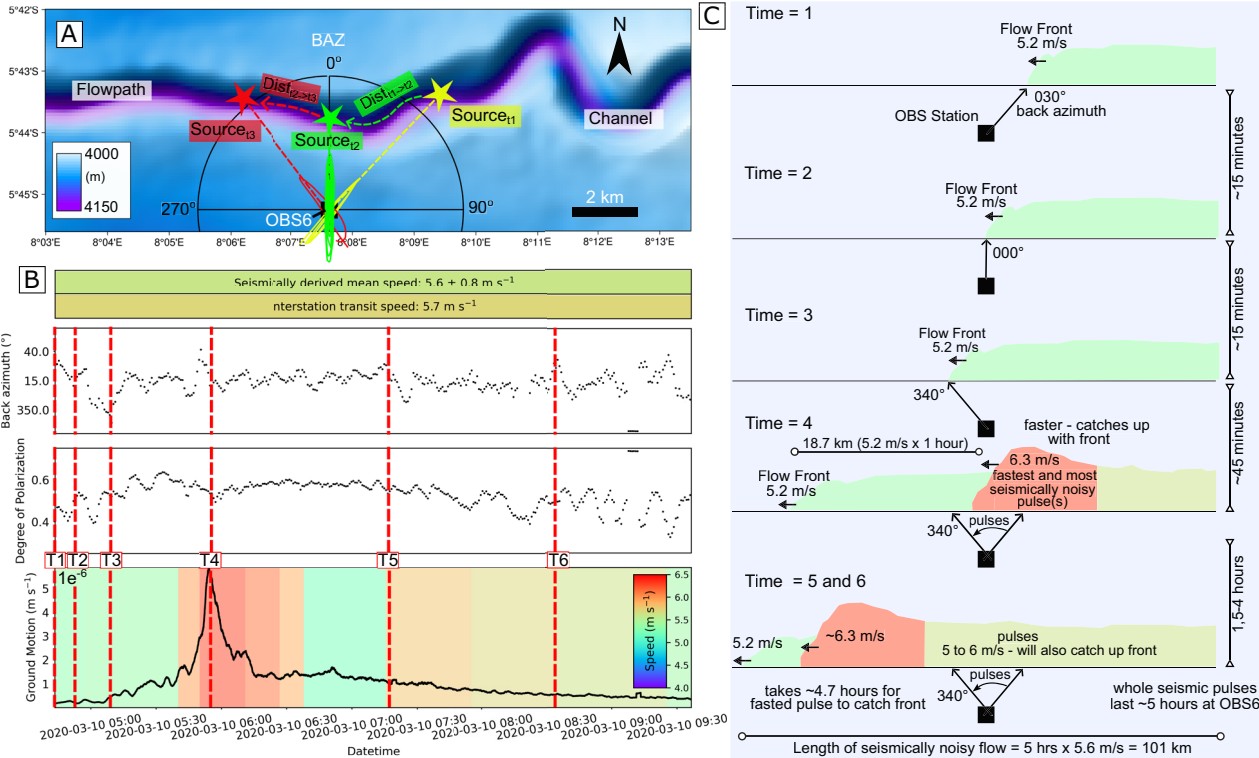

**Fig. 4 | Analysis of seismic signals from the turbidity current on 8th March 2020 recorded at OBS 6. A** Map showing the location of OBS6 (black square), and geometry of the adjacent deep-water channel. The location of signal sources is determined by analyzing the polarization of Scholte waves emitted by the turbidity current, which provides the back-azimuth (BAZ) direction. The source location is inferred to be where this back-azimuth direction intersects the canyon axis, such as at time $t_1$ (yellow lines), $t_2$ (green lines) and $t_3$ (red lines). Changes in source location through time are used to determine the velocity at which this dominant source moves down canyon. For instance, the distance moved along the canyon axis is divided by the difference in time between $t_1$ and $t_2$, or $t_2$ and $t_3$. **B** (upper panel) Time series BAZ direction, measured clockwise from north (0° or 360°), indicates multiple sources (surges) that move down the channel. For each surge, the BAZ direction is from 40°-20° as the surge approaches the OBS station, rotates to 10° as it passes the

nearest point to the OBS, and then decreases below 350° as the surge moves away from the OBS station. Time-lines (vertical red lines; times = 1 to 6) refer to snap shots of flow structure subsequently depicted within Figure **C**. (middle panel) The Degree of Polarization (DOP) corresponding to these BAZ estimations shows high values during the flow, indicating stable and reliable signal polarization for most of the signal (>0.4). (lower panel) Ground motions of the vertical component of the seismic waveforms recorded at OBS 6 for the March 8th 2020 turbidity current showing the derived speed structure in the colored background. The interstation transit speed and seismically derived mean speed plotted above as a bar. **C** Snapshots of the flow structure reveal that the leading edge of the flow exhibited slower speeds compared to the trailing surges. The fastest surge occurred approximately 18.7 kilometers behind the leading edge and was 21% faster. These surges lasted for approximately five hours.

We can also track how surge velocities change over time as they pass fixed OBS positions, such as OBS6 (Fig. 4). In the first 30–60 minutes, surges have lower speeds, averaging 5.2 m s⁻¹ (Fig. 4C). About an hour after the flow front passes, faster surges reach OBS6 with average speeds of 6.3 m s⁻¹. This time delay of roughly one hour implies the fastest surges occur roughly 19 km behind the front, assuming the front maintains a speed of 5.2 m s⁻¹. Surge speeds then decrease to 5–6 m s⁻¹ for the next three hours of the seismically noisy part of the flow. The later part of the flow did not produce clear polarized Scholte waves (Fig. S3), making the source trajectories from the seismic station unclear and leaving their speeds unknown. However, elevated water temperatures suggest the flow may have continued for days or weeks[33].

The overall seismic waveform—an emergent arrival, a maximum, and a long decay—is not due to the flow front's approach, peak proximity to the station, and subsequent distancing. The flow front passes the OBS6 30–60 minutes before the peak seismic energy, which rather corresponds to the fastest surges. These faster-moving surges likely carry higher sediment concentrations, contributing to their increased speeds. This pattern aligns with observations of debris flows, where coarse surge fronts generate stronger seismic amplitudes than the later, slower parts of the flows[39]. Consequently, this also explains why faster surges produce stronger seismic signals than slower surges, even when these surges come from the same back azimuth and thus position along the canyon-axis. This observation indicates

that the highest sediment concentrations and fastest surges within canyon-flushing turbidity currents are located a considerable distance (e.g. 19 km) behind the flow front.

## How do pulses originate?

Turbidity currents in the Congo Canyon exhibit pulsing behavior across different spatial and temporal scales, likely driven by multiple processes. In the upper canyon, flows with long-duration pulses (2–6 hours) have been observed[33] (Fig. S4), which are thought to originate from multiple upstream events at the Congo River mouth (e.g. triggered by spring tides[5]) or from multiple landslides along the canyon's walls in its first 100 km. These pulses traveled downstream at different speeds, coalescing into a single pulse by the time they reached the deep-sea Congo Channel. Such pulse-amalgamation has also been documented in the Var Canyon in the Mediterranean Sea[16] and in laboratory experiments[40].

However, here we also observe shorter-duration pulses (5–20 minutes) that persist even further downstream in the Congo Channel (Figs. 4, S5). These shorter pulses may have two possible origins. One possibility is that these shorter pulses are caused by "external" processes, such as localized erosion and entrainment of seabed sediment, which create denser and faster surges inside the flow. This process is similar to that observed in snow avalanches, where the initial movement triggers further failures along the margins of the avalanche track[26]. This model is consistent with patchy

erosion by these turbidity currents in the Congo Canyon and Channel[5,35] and suggests that some surges originated via failures triggered 30–60 minutes (or 9–19 km) behind the flow front.

Alternatively, surges may arise "internally" from small initial perturbations that grow over time[25,41]. For example, instabilities (called roll waves) form in thin and fast flows of water when the flow becomes supercritical, with Froude number exceeding one[25,42]. Surges also form via internal processes within high-sediment concentration flows. For example, surges are ubiquitous within subaerial debris flows, both in the field and large-scale experiments[23,25]. These debris flow surges arise from slight variations in grain size distribution, which affect granular friction and pore pressure, influencing flow speed and discharge, and amplifying initial disturbances[23,25]. Debris flow surges also amalgamate as they runout at different speeds, growing in size and duration. Similar pulsing also occurs in dry granular flows, arising via local grain entrainment from the underlying bed in erosion-deposition waves[43].

This 'internal' model is favored here as it better explains the quasi-uniform spacing of surges in the Congo Canyon, which is consistently seen at multiple OBS sites and in multiple turbidity currents (Fig. S5-6). In contrast the 'external' model would likely produce surges that were more randomly spaced and less consistent across different flows.

## Pulses sustain flow front

The observed differences in surge speeds suggest that faster-moving surges tend to catch up with slower-moving flow fronts (Fig. 4C), supplying additional sediment and momentum. This process may play a key role in driving the turbidity current front downslope and sustaining its exceptionally long runout distance (>1000 km; Fig. 5).

Traditionally, turbidity current front speeds have been modeled based on a local balance between gravitational driving forces (related to flow thickness, excess density, sediment concentration, and seabed gradient) and frictional forces at the flow front[10]. However, our findings suggest that this perspective omits a critical factor: momentum transfer from within the flow itself. Specifically, faster-moving, higher sediment concentration 'internal surges' can deliver additional momentum to the flow front. This mechanism

implies that front speeds may be influenced not only by local force balances but also by the internal structure and dynamics of the turbidity current.

The transfer of sediment into the flow front via surges may also explain why the Congo Canyon flushing flows sustained the near-uniform front speeds over hundreds of kilometers despite prodigious erosion[5,33], by counterbalancing the loss of sediment through mixing with surrounding seawater, deposition, or other processes.

## Why are the fastest pulses so far behind the flow front?

It is expected that the front of a turbidity current is slower than the body, as it experiences greater friction while displacing surrounding seawater, partly due to more vigorous mixing near the front. In small-scale laboratory experiments, turbidity currents typically have a front speed that is ~30–40% lower than the body's speed[10], consistent with our observations from the Congo Canyon where the front speed is 20–40% slower than the maximum internal surge speed (Figs. 4, S5–6).

However, in laboratory experiments, the maximum speed occurs just a few seconds behind the flow front, corresponding to a distance of only a couple of flow thicknesses. In contrast, the maximum surge speeds in these canyon-flushing flows occur much further behind the front, with a time lag of 30–60 minutes. Assuming a front speed of ~5.2 m s$^{-1}$, this time lag equates to a distance of ~9–19 km. Given a flow thickness of ~100–120 m (channel is 100 m deep at OBS6[44]), this distance of 9–19 km corresponds to 75–190 times the flow thickness. Even in previously monitored slower (<2 m s$^{-1}$) Congo Canyon flows[12], the maximum speed occurs 10–30 minutes after the flow front—about 1.2 km behind the front—equivalent to ~60 flow thicknesses behind the front.

This indicates that maximum flow speeds occur much further behind the front in Congo Canyon turbidity currents than in laboratory flows, suggesting substantial differences between field-scale and laboratory flows, including potentially greater frontal mixing and friction in the Congo Canyon flows.

These unique field observations underpin a new view of the internal structure and dynamics of exceptionally large canyon-flushing turbidity currents (Fig. 5). Canyon-flushing turbidity currents consist of a series of

**Fig. 5 | Surge dynamics of canyon-flushing turbidity currents.** Canyon-flushing turbidity currents consist of surges lasting 5–30 minutes, persisting for many hours and extending tens to hundreds of kilometers. The fastest surges, which carry the highest sediment concentration, occur 30–60 minutes behind the flow front and can be as far as 20 km back(t0). These higher sediment concentration pulses eventually overtake the front (t1), playing a key role in supplying sediment and momentum to sustain the exceptionally long runout distances (>1000 km) of these seafloor flows.

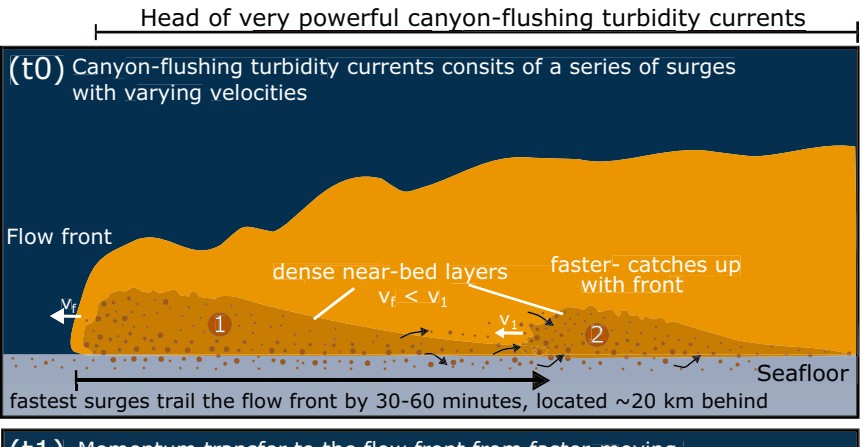

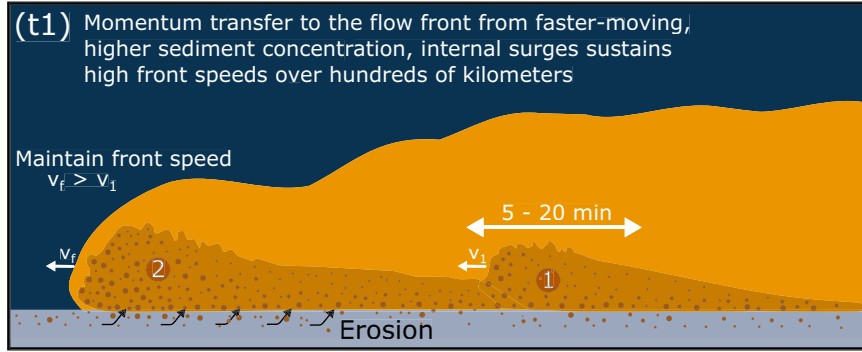

pulses lasting 5–20 minutes, with the fastest pulses occurring 30–60 minutes behind the flow front. These fast pulses generate the strongest seismic signals, indicating higher sediment concentrations, and play a crucial role in transferring momentum to the flow front, driving the current forward over exceptionally long distances. The pulses are likely generated spontaneously through the amplification of small initial perturbations, similar to mechanisms observed in subaerial debris flows[23,25]. The widespread presence of these pulses suggests a high sediment concentration at the base of the flow.

The occurrence of surges has important implications for the magnitude and duration of impact forces on seabed cables, explaining why these cables often break. As seen with snow avalanches[26], the maximum speed of surges can be much higher than that of the flow front. Calculating impact forces based on frontal speeds may, therefore, significantly underestimate the peak loading that cables experience.

## Methods

### Seismic data collection
Eleven ocean-bottom seismometers (OBS) were deployed along the Congo Submarine Canyon and Channel in September-October 2019, organized into canyon and channel sub-arrays (Fig. 1). Nine of the OBSs were recovered, and recorded data for approximately 9–10 months. Each OBS was equipped with three-channel Sercel L28-LB geophones, positioned 600–2900 meters from the center of the canyon-channel, on flat canyon terraces or overbank areas outside the channel. The sampling frequency was 1 kHz for all stations, except for OBS 3, which sampled at 250 Hz. The corresponding seismic traces were detrended, and instrument responses were removed to obtain velocity units (m s$^{-1}$) using the open-source Python framework ObsPy[45]. The data were band-pass filtered between 0.02 and 25 Hz, then down-sampled to a rate of 50 Hz (Nyquist frequency of 25 Hz). The horizontal components of the OBS seismometers were oriented to geographic north and east, with their alignment derived using the polarization characteristics of seismic waves[46,47] from independently located earthquakes. The estimated uncertainty in this alignment is ±11.5 degrees.

### Spectrograms and waveform envelopes
Spectrograms are generated using Welch's[48] method with 300-second windows and 50% overlap. The probability spectral density (PSD) is presented in decibels (dB), relative to velocity ($10*\log_{10}[(m^2 \, s^{-2}) \, Hz^{-1}]$). Turbidity currents are visually identified on OBS spectrograms by distinct high-frequency vertical ground motions, with predominant power between 0.5 and 10 Hz. Signal-to-noise ratios for these events vary between 5 and 30 dB above ambient noise (Fig. S7). Lower ratios are primarily observed in the upper canyon region, influenced by seismic noise from anthropogenic sources such as shipping and drilling, which also generate high-frequency noise.

The amplitude of turbidity current signals is represented by the envelope of high-frequency vertical ground motions (filtered between 0.5 and 10 Hz) and smoothed with a 1-minute sliding window. The waveform onset shows a gradual increase in both frequency and amplitude (Fig. 2), tapering off before after peak values. High-frequency components diminish first, followed by a gradual reduction in lower frequencies. The detection of turbidity currents by the OBS is further validated by ADCP-mooring measurements, identified by an abrupt increase in near-bed velocity above background levels[33].

### Seismic characterization of turbidity currents
Once a seismic signal is identified as a turbidity current, it is further characterized. This process is illustrated using data from OBS6 for the March 8th, 2020 event (Fig. 4), and involves several stages:

Stage 1: Polarization analysis. We conducted a frequency-dependent polarization analysis[49] on the three-component turbidity current signals to determine polarization attributes, namely the degree of polarization (DOP), vertical to horizontal phase difference (phiVH), and back azimuth (BAZ) across the 1–10 Hz frequency band. The approach follows established methods from ambient noise studies in various environments that have characterized particle motions from natural sources[50–52]. In our analysis, we use 150-second sub-windows with 50% overlap to compute polarization attributes, allowing detailed characterization of the evolving signal sources.

The back azimuth (BAZ) represents the trajectory of incoming seismic waves, measured in the horizontal plane and expressed in degrees relative to true north. It provides key information about the signal source's location relative to the seismic station.

The vertical to horizontal phase difference (phiVH) characterizes the phase relationship between the vertical and horizontal components of the seismic signal, with values bounded between −90° and 90°. We analyze the absolute phase angle difference, as the magnitude of this shift provides a clearer interpretation of polarization, focusing on the extent of the phase shift rather than its direction—whether the horizontal component leads or lags the vertical. This helps in identifying the type of seismic waves, as different wave types exhibit distinct phase relationships between the vertical and horizontal components (see Stage 2).

The Degree of Polarization (DOP) ranges from 0 to 1, with 0 indicating unpolarized or isotropic wave motion, and values near 1 suggesting more linear or elliptical polarization[53]. A higher DOP reflects a wavefield dominated by a single component, indicating a coherently propagating seismic wave with greater directionality. This metric can therefore be used to gauge confidence in the derived BAZ.

Stage 2: Wave type and quality assigned. Seismic signals must exhibit sufficient polarization and Scholte wave-like characteristics to derive their back azimuths. Scholte waves are typically identified by their 90-degree phase lag, and we classify signals with a phase lag between 70° and 90° as Scholte wave-like. A phase lag between 30° and 70° suggests a mix of wave types, such as a combination of body and surface waves. Phase lags near 0° are often associated with body waves and are excluded from further analysis.

A signal's arrival is defined by a coherent increase in energy, exhibiting the expected polarization and phase shift characteristics of Scholte waves, distinctly different from background seismic signals, which typically show lower amplitude, random polarization, and unclear phase relationships. Our focus is on characterizing the dominant signal sources from turbidity currents, rather than capturing particle motion from all seismic energy sources. We identified sufficiently polarized Scholte waves emitted by the turbidity currents, typically in the 4–6 Hz band (Fig. S3).

Stage 3: Evolving location and speed of turbidity current signal sources along the canyon-channel. To determine the location of the dominant turbidity current signal source along the sinuous canyon-channel axis and track its evolution over time, we projected BAZ trajectories derived from sufficiently polarized high-frequency Scholte waves onto the line of steepest descent along the canyon axis, assuming the turbidity current remains naturally confined to this path.

From these projections, we calculated the speed of the dominant source along the canyon axis. Additionally, oscillations in the polarization attribute direction were observed, which we infer to represent the propagation of multiple flow pulses. As one pulse moved away from the receiver, the back azimuth gradually veered down-flow. A subsequent, stronger pulse from up-flow caused the back azimuth to shift back toward the up-flow direction, indicating the arrival of the new dominant source.

This analysis also helped define the effective detection range of seismic stations for turbidity currents, allowing us to determine the distance at which these flows can be detected and located by seismic sensors, within canyon segments approximately 3–7 km from the measurement location.

### Models of seismic signal generation from turbulent flow and sediment transport
To investigate whether the seismic signals from turbidity currents arise from flow turbulence, sediment transport within the flow, or both, we analyze seismic records from stations at varying distances from the canyon. This allows us to distinguish signal sources based on differences in their spectral signatures[29]. We then compare these signatures to those predicted by models of turbulent flow[36] and bedload transport[37] from river systems, applied here to the marine environment, to inform interpretation of potential signal sources.

Both models are physically based and capture the first-order processes generating ground motion detectable by seismic stations. The bedload transport model[37] links seismic noise generation to sediment transport, where individual particles impacting the riverbed create force impulses that generate Rayleigh waves. These impacts, modeled as elastic contact problems, occur at random intervals and are influenced by factors like grain size and flow conditions. The model computes the total seismic noise power spectral density (PSD) by summing these impacts and assumes that the sediment transport rate determines the frequency of impacts, with seismic energy proportional to sediment flux.

The turbulent flow model[36] attributes seismic noise to fluctuating forces from turbulent water flow. Pressure and shear stresses exerted on the riverbed by turbulence cause vibrations that propagate as Rayleigh waves, detectable by nearby seismic stations. The magnitude and characteristics of these forces depend on key hydraulic parameters, including water depth, bed roughness, and flow velocity—where flow velocity itself is influenced by the river's slope, channel roughness, and flow depth. The model treats these forces as stochastic processes to reflect the inherent randomness of turbulence. By integrating the contributions of these forces over a range of frequencies, the model calculates the Power Spectral Density (PSD) of the resulting seismic noise, establishing a quantitative link between river flow properties and their seismic signatures.

Implementing these models requires setting or estimating various parameters, including fluid properties, sediment characteristics, bed conditions, and the medium for seismic wave propagation. Due to the limited constraints on certain sediment transport parameters, particularly sediment flux, for Congo Canyon turbidity currents[5,12], we relied on empirical data and best estimates from terrestrial studies to address these gaps.

We set the flow height to 50 m and sediment flux to 0.4 m² s⁻¹. The specific sediment density was defined as 2650 kg m⁻³, the fluid density as 1024 kg m⁻³, and the canyon slope as 0.5 degrees. A log-normal, raised-cosine grain size distribution[37] was used, with a mean grain diameter of 0.02 m and a standard deviation of 0.5. The ground quality factor was set to 40, and the Rayleigh phase wave velocity to 300 m s⁻¹.

We modeled source-to-station distances of 800 and 1600 m to reflect seismic data from OBS2 and OBS3, respectively. Assuming consistent conditions within this channel segment, all parameters were kept constant except for the distance from the source. Model spectra were then generated for sediment transport and flow turbulence, both individually and combined (Fig. S1).

Understanding the seismic signals generated by turbidity currents requires reconciling key differences between theoretical models and the complexities of real-world flows. For instance, while the turbulent flow[36] and bedload transport[37] models assume Rayleigh waves without a water layer, Scholte waves are present in our marine setting. This discrepancy could introduce slight differences in waveform characteristics, such as seismic amplitude. However, these differences are not expected to fundamentally alter the qualitative comparison of spectral shapes and trends.

Additionally, the Congo Canyon turbidity currents likely featured a wide range of grain sizes and potentially contained near-bed layers with much higher (>20–40% volume)[5,12] sediment concentrations than rivers. These dense near-bed layers may resemble debris flows or hyperconcentrated flows more than typical bedload fluvial transport, leading to deviations from the models' assumptions. For example, grain collisions in higher sediment concentration flows may be buffered by elevated pore pressures and exhibit prolonged, inelastic behavior[23,54,55].

Given these complexities, our approach focused not on achieving precise quantitative matches between seismic observations and model outputs, but on highlighting key processes. Specifically, we sought to illustrate how sediment transport and turbulence distinctly influence seismic signal strength and frequency range as the distance from canyon axis to receiver increases. This comparison supports the idea that two distinct processes—sediment transport and flow turbulence—generate seismic signals in turbidity currents.

Sediment transport in turbidity currents is more complex than these models suggest, involving processes beyond perfectly elastic collisions. We hope this analysis encourages future research to better understand sediment transport in turbidity currents and the nature of these flows.

## Data availability

The raw OBS seismic data used to record turbidity currents in this study are available at the British Oceanographic Data Centre (BODC) with no access conditions (https://doi.org/10.5285/1D4A3BF0-FEAB-465C-E063-7086ABC0EF74)[56].

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

## Acknowledgements

We gratefully acknowledge the shipboard parties of the RRS James Cook cruises JC187 and JC209. Our thanks to GEOMAR and NERC's Ocean-Bottom Instrumentation Facility (OBIF) for providing OBSs, and to Angola Cables for organizing permissions for work in Angolan waters. Funding for P.K. was provided by the Deutsche Forschungsgemeinschaft (DFG, Grant No. 504509409). M.U. has received funding from the European Research Council (ERC) under the European Union's Horizon 2020 research and innovation program (grant agreement No. 948797). This research was supported by the National Environment Research Council grants NE/R001952/1, NE/S010068/1, and NE/R015953/1. M.L.B. received funding through a Leverhulme Trust Early Career Fellowship, ECF-2021-566.

## Author contributions

Conceptualization: P.K. Data Acquisition: P.J.T., M.L.B. Methodology: P.K., D.L., W.-C.C. Data Analysis: P.K. P.K. drafted the paper, while P.J.T., D.L., W.-C.C., M.L.B., M.U., and C.B. contributed to discussions of the dataset and provided comments and corrections to the manuscript.

## Funding

## Competing interests

The authors declare no competing interests.
