## [Peer review file · Communications Earth & Environment]

Ocean-bottom seismometers reveal surge dynamics in Earth's longest-runout sediment flows

Corresponding Author: Dr Pascal Kunath

Version 0:

Decision Letter:

Dear Dr. Kunath,

Your manuscript titled "Ocean-Bottom Seismometers Reveal Surge Dynamics in Earth's Longest-Runout Sediment Flows" has now been seen by 3 reviewers, and we include their comments at the end of this message. They find your work of interest, but some important points are raised. We are interested in the possibility of publishing your study in Communications Earth & Environment, but would like to consider your responses to these concerns and assess a revised manuscript before we make a final decision on publication.

Besides all reviewers' comments, you may want to take into account the possibility of comparison with signals from debris flow dynamics.

We therefore invite you to revise and resubmit your manuscript, along with a point-by-point response that takes into account the points raised. Please highlight all changes in the manuscript text file.

Please submit your point-by-point responses as a separate file, distinct from your cover letter where you can add responses to the Editors' comments that you do not want to be made available to the reviewers. Word files are preferred. We recommend that any figures, tables or graphs that are included in the response to reviewers are also included in the main article or Supplementary Information.

Please use the following link to submit your revised manuscript, point-by-point response to the referees' comments (which should be in a separate document to any cover letter), a tracked-changes version of the manuscript (as a PDF file) and the completed checklist:

Link Redacted

We hope to receive your revised paper within six weeks; please let us know if you aren't able to submit it within this time so that we can discuss how best to proceed. If we don't hear from you, and the revision process takes significantly longer, we may close your file. In this event, we will still be happy to reconsider your paper at a later date, as long as nothing similar has been accepted for publication at Communications Earth & Environment or published elsewhere in the meantime.

Please do not hesitate to contact us if you have any questions or would like to discuss these revisions further. We look forward to seeing the revised manuscript and thank you for the opportunity to review your work.

Best regards,

Domenico M. Doronzo, PhD

Editorial Board Member
Communications Earth & Environment
orcid.org/0000-0002-6866-8870

Joe Aslin
Deputy Editor
Communications Earth & Environment

EDITORIAL POLICIES AND FORMATTING

Editorial Policy: [Policy requirements](https://www.nature.com/documents/nr-editorial-policy-checklist.pdf) (Download the link to your computer as a PDF.)

- Behavioural and social science
- Ecological, evolutionary & environmental sciences
- Life sciences

<https://www.nature.com/documents/nr-reporting-summary.zip>

Furthermore, please align your manuscript with our format requirements, which are summarized on the following checklist: [Communications Earth & Environment formatting checklist](https://www.nature.com/documents/commsj-phys-style-formatting-checklist-article.pdf)

and also in our style and formatting guide [Communications Earth & Environment formatting guide](https://www.nature.com/documents/commsj-phys-style-formatting-guide-accept.pdf).

*** DATA: Communications Earth & Environment endorses the principles of the Enabling FAIR data project (<http://www.copdess.org/enabling-fair-data-project/>). We ask authors to make the data that support their conclusions available in permanent, publically accessible data repositories. (Please contact the editor if you are unable to make your data available).

All Communications Earth & Environment manuscripts must include a section titled "Data Availability" at the end of the Methods section or main text (if no Methods). More information on this policy, is available at <http://www.nature.com/authors/policies/data/data-availability-statements-data-citations.pdf>.

If a community resource is unavailable, data can be submitted to generalist repositories such as [figshare](https://figshare.com/) or [Dryad Digital Repository](http://datadryad.org/). Please provide a unique identifier for the data (for example a DOI or a permanent URL) in the data availability statement, if possible. If the repository does not provide identifiers, we encourage authors to supply the search terms that will return the data. For data that have been obtained from publically available sources, please provide a URL and the specific data product name in the data availability statement. Data with a DOI should be further cited in the methods reference section.

REVIEWER COMMENTS:

Reviewer #1 (Remarks to the Author):

This submission seeks to utilize novel techniques for monitoring turbidity currents with Ocean Bottom Seismometers, calibrated with in-situ velocity data from ADCP's and physics-based models of the seismicity of sediment transport processes from studies in riverine sediment transport. The manuscript is expertly written, the data of extremely high quality, and the analyses are novel and robust. I recommend acceptance without revision.

One question from Table 1, however: Shouldn't the units of sediment flux be m^3/s , not m^2/s ? Also, not that it makes a big difference, but why wouldn't one use a water density closer to that of seawater rather than freshwater for input (i.e., ~ 1024 , not 1000), as I believe that factors into the viscosity, does it not?

Reviewer #2 (Remarks to the Author):

This is a very polished article describing an analysis that is unique as far as I am aware of OBS recordings of turbidity currents (or hybrid flows?). The separate geophones in the OBSs allow the direction of propagation of the waves to be worked out. The results suggest that the flows contain surges and documents their frequency and speeds. The technical developments promise more exciting work in this area, if OBSs can be deployed at other canyons and in greater numbers.

I had few comments to make. Some readers might question why OBSs are needed and not simply hydrophones (I believe hydrophones have been used previously though not so effectively, I think in the Monterey Canyon). The direction of water-borne waves would be impossible to detect with single hydrophones. Also, the speed of sound in water increases with depth below the thermocline, so waves refract upwards, which limits the locations and range that hydrophone data can be useful over. I did wonder however what the hydrophone signals from these OBSs looked like and if they might help inform the use of hydrophones, as hydrophone data are more common.

Caplan-Auerback et al. described hydrophone recordings of submarine slope failures off Hawaii - I wondered if comparison with their results could be made and be useful? Their flows would have been more like debris flows than turbidity currents.

Minor suggestions

Line #

A minor issue, turbidity currents may be important in entrenching canyons, though they are not the only cause of canyon relief that has been proposed. Thalweg non-deposition can create canyon relief, while the inter-canyon ridges aggrade with accumulation of sediment. TCs can be involved in some of those alternative explanations.

"grave" sounds a bit too ominous to me, as few human lives are lost.

56 The word "incised" is loaded, when I think you really mean "relief". By incised, the text ignores the role of aggradation of areas outside the channel.

71 km^3 ? Superscript disappeared in this version.

text in brackets can be shortened to: (located in inset)

turbidity

between 1 and 7

"signals" needs to be more specific - I think you mean ground motion. I see you use this later on as well - perhaps a brief clarification is needed if you are referring to both amplitude and PSD.

The cigar shape wasn't obvious at first - perhaps refer to Figure S4?

95 exponential forms may need to be in a different format for publication, ie 10^{-7}

98 OBSs

P-wave attenuation is more power-law, so presumably it is quasi-linear because of the frequency difference is small.

It is unclear why you are referring to Figure 3c here.

(D,E)

208 passes -> passing (?)

378 This allows us to

401 You don't need phrases like "it is important to note that". These make the text harder to read as the information becomes more dispersed.

415. It is not clear what log m means for the units. Presumably $\log_{10}(\sigma)$ - please also verify the base of the logarithm.

Supplementary file:

absolute -> magnitude

99 Deploying the OBSs on the terraces outside the thalweg presumably ensures the Scholte wave propagation speed is fairly uniform spatially and there is limited effect of refraction? Elsewhere, there might be variations in thickness of the surface sediment layers that might lead to spatial varied wave speed?

A very easy and enjoyable read, I look forward to seeing it in "print"

Neil Mitchell

University of Manchester, November 2024

Caplan-Auerback, J., Fox, C.G. and Duennebieber, F.K., 2001. Hydroacoustic detection of submarine landslides on Kilauea volcano. *Geophys. Res. Lett.*, 28: 1811-1813.

One or both of these included marine Scholte waves, in case useful:

Stoll, R.D., 1985. Marine sediment acoustics. *J. Acoust. Soc. Am.*, 77: 1789-1799.

Stoll, R.D., 1991. Geoacoustic properties of a marine silt. In: R.H. Bennett, W.R. Bryant and M.H. Hulbert (Editors), Chapter 43, Microstructure of fine-grained sediments. Springer-Verlag, New York, pp. 395-402.

Reviewer #3 (Remarks to the Author):

Review of "Ocean-Bottom Seismometers Reveal Surge Dynamics in Earth's Longest-Runout Sediment Flows" by Kunatg et al., submitted to Nature Earth and Environment

GENERAL COMMENTS

In the present paper the authors provide unprecedented observations of sea-floor turbidity currents from seismic monitoring. The report particularly insightful observational features. Mainly, they are able (i) to dissociate the turbulence versus sediment transport seismic signatures based on evaluating seismic power variations with distance, (ii) to identify that the turbidity current operates in 5-30 min long pulses propagating at varying speeds, and that (iii) the most energetic part of the turbidity current is not concentrated in the front but rather farther up the flow. These results have implications on the physics controlling the formation and propagation of these events, which are discussed in the paper. Overall, the seismic data analysis is thorough, very sound and particularly convincing. The paper is well written and addresses a particularly interesting and yet poorly known Earth mechanism. For all these reasons I am strongly in favor of publication in Nature Earth and Environment.

I have a few general comments that I think the authors could use to improve the draft, in particular to make it better suited for Nature Earth and Environment, as well as more specific comments that I provide below.

GENERAL COMMENTS

The abstract and introduction are good, but I think there is a lack of hierarchy in introducing aspects related to the key findings of the authors, and how they have the potential to change our view on how turbidity current operates. At several instances in the abstract and the introduction it is hard to dissociate what was known before from what the present paper brings in. For example, the typical runout and velocities seem to be already known, such that the ability of seismic observations to constrain these aspects might not be the best-selling point to emphasize, or at least not without more details about the advantages offered by seismic observations (e.g. spatial and temporal scales). Instead, the surge behavior consisting in repetitive pulses seems to be a new finding with important implications, which appears to me as comparatively under sold, i.e. under introduced in the abstract and intro, and not sufficiently discussed in light of existing literature towards the end of the paper. I find the discussion line 253-261 on this aspect to be difficult to follow, although this point seems quite important for the broader implications of the current findings. In addition, I think there is lacking information regarding our knowledge on the seismic signature of mass movements on earth (rivers, debris flows) and to which extent one might think to extend this to turbidity currents. There should be at least a short paragraph on this in the introduction and some extended discussion later on.

SPECIFIC COMMENTS

Line 63: Reference to an EGU presentation to argue something fundamentally important is I think not appropriate. The authors claim that "the seismic signals originate from the faster-moving frontal zone" as an already demonstrated finding, but the citation has not been peer-reviewed, thus I don't think it is appropriate.

Line 77 to 81 : The list of study objectives is quite complete, but it is hard to connect them with the various lack of knowledge discussed earlier. I would suggest to make a better connection, for example by disseminating the objectives after each lacking knowledge paragraphs.

Figure 3 : really interesting analysis. Would be nice to have the observed spectrum of OBS3-OBS2, to be more easily compared with the theoretical one. I assume the authors did not show it because maybe it did not readily work, but that is ok, the conceptual model and the fact that this feature exists in the data is convincing, it would just be easier for the reader to evaluate the degree up to which it is indeed convincing.

Line 375: - The models of Gimbert al and Tsai et al. have been developed for surface waves with no water layer on top. In the present case the water layer has to be accounted for, which I think is done by the author, based on the reference to Scholte wave in Table 1. However, I think much more information is required on how the authors describe this Scholte wave, and how seismic wave amplitude may differ in this case compared to the classical surface wave case investigated previously.

My comment above regarding lacking information on the Green function actually also applies to the seismic source description of turbidity current. In particular, the crucial parameter in the theory of Gimbert et al. is the water flow velocity. The authors must be using a description of water flow velocity as a function of flow height and surface slope, but they are not explaining this. There is also a lot of complementary information missing about the application of the model to their present setup. I think this information must be provided.

Line 152 : reference to this statement ? I have a hard time understanding what these sentences relate to, previous work or current one, if the later then refer to figure ?

Line 224 : remove "into"

L262: You might consider citing Cook et al., 2021, who observed a river flood also with seismics constituted by a flow front that's less energetic than a later arriving flow component, which the authors attributed to a more concentrated sediment wave. In that case, however, the second arriving flow is slower than the front, which is different than observed here, but maybe tells us something about the similitudes/differences in the underlying physics ?

Communications Earth & Environment is committed to improving transparency in authorship. As part of our efforts in this direction, we are now requesting that all authors identified as 'corresponding author' create and link their Open Researcher and Contributor Identifier (ORCID) with their account on the Manuscript Tracking System prior to acceptance. ORCID helps the scientific community achieve unambiguous attribution of all scholarly contributions. You can create and link your ORCID from the home page of the Manuscript Tracking System by clicking on 'Modify my Springer Nature account' and following the instructions in the link below. Please also inform all co-authors that they can add their ORCIDs to their accounts and that they must do so prior to acceptance.

Version 1:

Decision Letter:

Dear Dr Kunath,

Your manuscript titled "Ocean-Bottom Seismometers Reveal Surge Dynamics in Earth's Longest-Runout Sediment Flows" has now been seen by our reviewers, whose comments appear below. In light of their advice we are delighted to say that we are happy, in principle, to publish a suitably revised version in Communications Earth & Environment.

We therefore invite you to edit your manuscript to comply with our format requirements and to maximise the accessibility and therefore the impact of your work.

EDITORIAL REQUESTS:

*****Please take care to match our formatting and policy requirements. We will check revised manuscript and return manuscripts that do not comply. Such requests will lead to delays. *****

SUBMISSION INFORMATION:

OPEN ACCESS:

Communications Earth & Environment is a fully open access journal. Articles are made freely accessible on publication. For further information about article processing charges, open access funding, and advice and support from Nature Research, please visit <https://www.nature.com/commsenv/open-access>

Link Redacted

Best regards,

Domenico Doronzo
Editorial Board Member
Communications Earth & Environment

Joe Aslin
Deputy Editor,
Communications Earth & Environment
<https://www.nature.com/commsenv/>
Twitter: @CommsEarth

REVIEWERS' COMMENTS:

Reviewer #2 (Remarks to the Author):

Pascal and co-authors have done a thorough job of working through the comments and updating the manuscript where necessary. I would quibble a little with the response to my comment on the earlier MS line 99. Without knowing so much about Scholte waves, P-waves vary by 2-3X or more between rock and unconsolidated sediments, so in other settings (rock outcrops with varied thickness sediment) I would expect refraction (ray-path bending). Nevertheless, this does not appear to be an issue for the present study area and the uniform velocity assumption should be obvious.

Reviewer #3 (Remarks to the Author):

I am fully satisfied by the answers and modifications provided by the authors, which I think improve the manuscript. I highly recommend this contribution for publication.

Florent Gimbert

**Reply to Reviewers**

We thank the reviewers for their constructive and detailed feedback, which has significantly improved
our manuscript. Below, we provide our responses to each comment in red text, accompanied by quoted
passages from the revised or newly added text in the manuscript, formatted in italics and underlined.
Line references correspond to the revised manuscript without tracked changes.

Reviewer #1

This submission seeks to utilize novel techniques for monitoring turbidity currents with Ocean Bottom
Seismometers, calibrated with in-situ velocity data from ADCP's and physics-based models of the
seismicity of sediment transport processes from studies in riverine sediment transport. The manuscript
is expertly written, the data of extremely high quality, and the analyses are novel and robust. I
recommend acceptance without revision.

We thank Reviewer 1 for their positive assessment of our manuscript.

One question from Table 1, however: Shouldn't the units of sediment flux be m^3/s , not m^2/s ? Also,
not that it makes a big difference, but why wouldn't one use a water density closer to that of seawater
rather than freshwater for input (i.e., ~ 1024 , not 1000), as I believe that factors into the viscosity, does
it not?

The units represent a volumetric flux per unit width, which explains the units of m^2/s . To obtain the total
volumetric flux (m^3/s), this per-unit-width flux can be multiplied by the channel width.

Regarding the fluid density, we thank you for catching that oversight. We have corrected the typo, as
we had indeed used a density closer to that of seawater ($\sim 1024 \text{ kg/m}^3$) in our calculations.

Reviewer #2

This is a very polished article describing an analysis that is unique as far as I am aware of OBS
recordings of turbidity currents (or hybrid flows?). The separate geophones in the OBSs allow the
direction of propagation of the waves to be worked out. The results suggest that the flows contain surges
and documents their frequency and speeds. The technical developments promise more exciting work in
this area, if OBSs can be deployed at other canyons and in greater numbers.

I had few comments to make. Some readers might question why OBSs are needed and not simply
hydrophones (I believe hydrophones have been used previously though not so effectively, I think in the
Monterey Canyon). The direction of water-borne waves would be impossible to detect with single
hydrophones. Also, the speed of sound in water increases with depth below the thermocline, so waves
refract upwards, which limits the locations and range that hydrophone data can be useful over. I did
wonder however what the hydrophone signals from these OBSs looked like and if they might help
inform the use of hydrophones, as hydrophone data are more common.

We thank the reviewer for their insightful comments on the potential application of hydrophones for
detecting turbidity currents. Hydrophones have proven effective in capturing acoustic signals generated
by submarine mass movements in certain environments, as demonstrated in previous studies (e.g.,
Caplan-Auerbach et al., 2001). However, in our case, hydrophone records from the Congo Canyon did
not reveal any detectable acoustic signals, even from the very large canyon-flushing turbidity currents
studied here. For example, the figure reproduced below is taken from the Baker et al. (2024) published
in Geophysical Research Letters. This absence of a hydrophone signal suggests either a lack of
significant acoustic emissions from these flows, or conditions that inhibit the propagation of acoustic
emissions. The hydrophones on our seabed OBS lacked a direct line of sight to where the flows occurred
in the base of the entrenched canyon-channel, and this lack of a direct line of sight may be linked to the
lack of hydrophone signals. Whatever the exact reason, this study highlights the site-specific challenges
of using hydrophones for such monitoring.

We have added a short statement to this affect in the results section (line 108): Hydrophones mounted
on the OBSs failed to record the signals from the turbidity currents³³. However, we did not have space
to add another figure or expand discussion of the hydrophone data, and this is left for future contributions,
as it would complicate this paper's key messages.

Figure 1: Comparison of turbidity current signals recorded by the hydrophone and geophone on OBS6
from the large March 8th turbidity current event (reproduced from Supplementary Material S7 of Baker
et al. 2024). (a) Hydrophone spectrogram across the full frequency range and (b) hydrophone
spectrogram across the 1-25 Hz frequency range show that the hydrophone did not receive any acoustic

signals from the turbidity current. (c) Geophone spectrogram showing the seismic signals received from
the turbidity current.

Caplan-Auerback et al. described hydrophone recordings of submarine slope failures off Hawaii - I
wondered if comparison with their results could be made and be useful? Their flows would have been
more like debris flows than turbidity currents.

We appreciate the reviewer's suggestion to explore comparisons with the hydrophone recordings of
submarine slope failures discussed by Caplan-Auerbach et al. Their work highlights the potential of
acoustic methods for detecting mass movements; however, it does not delve into the detailed flow
processes that are central to our study. This limits the basis for direct comparisons between their
hydrophone-based observations and our geophone-based turbidity current measurements. At present,
we have limited information on how geophone and hydrophone signals from submarine mass
movements relate to each other or record the underlying flow processes. A logical next step would be
to deploy both instruments simultaneously and directly compare their signals. Indeed, we attempted such
an approach in the Congo Canyon but did not detect clear acoustic signatures associated with the
turbidity currents (see Figure 1 shown above). Further investigations are needed to determine why this
occurred and to establish a calibration framework for comparing different types of seismic and acoustic
data. For now, our study remains focused on geophone-derived insights.

We have included the reference to Caplan Auerbach in the introduction (line 56-57), as it also serves as
a good example of how seismic and/or acoustic methods can detect submarine mass flows. However,
compared to terrestrial remote sensing approaches for hazardous processes, its application to submarine
mass movements is still in its emerging phase.

Minor suggestions

Line #28 A minor issue, turbidity currents may be important in entrenching canyons, though they are
not the only cause of canyon relief that has been proposed. Thalweg non-deposition can create canyon
relief, while the inter-canyon ridges aggrade with accumulation of sediment. TCs can be involved in
some of those alternative explanations.

We agree that turbidity currents are not the sole drivers of canyon relief. To reflect this nuance, we have
revised the sentence as follows: Turbidity currents are the longest-runout sediment-driven flows on
Earth, playing a key role in shaping Earth's deepest and longest canyons and forming its largest
sediment accumulations. – (line 25)

33 "grave" sounds a bit too ominous to me, as few human lives are lost.

We have changed "grave threat" to "significant threat" – (line 30)

The word "incised" is loaded, when I think you really mean "relief". By incised, the text ignores the
role of aggradation of areas outside the channel.

We have revised this sentence accordingly: The submarine canyon exhibits significant relief (up to 1,200
91 m) along the continental shelf for the first ~150 km. It then transitions into a less pronounced deep-sea
channel (250–150 m deep) with depositional levees on its flanks, which terminates 1,100 km from the
river mouth at a depositional lobe. – (line 70)

71 km³? Superscript disappeared in this version.

We have corrected the superscript for km³ in the revised version of the manuscript.

text in brackets can be shortened to: (located in inset)

Done.

turbidity

Done.

between 1 and 7 "signals" needs to be more specific - I think you mean ground motion. I see you use
this later on as well - perhaps a brief clarification is needed if you are referring to both amplitude and
PSD.

Changed to "ground motions".

The cigar shape wasn't obvious at first - perhaps refer to Figure S4?

We have removed the statement that ground motion resembles a cigar shape and now describe it as: The
ground motions display emergent waveforms, characterized by a gradual build-up to a peak amplitude
later in the signal, followed by a subsequent gradual decay. – (line 105).

exponential forms may need to be in a different format for publication, ie 10⁻⁷

Corrected.

98 OBSs

Done.

P-wave attenuation is more power-law, so presumably it is quasi-linear because of the frequency
difference is small.

The term "quasi-linear" was not the most precise choice, especially given the nonlinear and frequency-
dependent nature of surface wave attenuation with distance. We have revised the phrasing for clarity,
focusing on the spectral differences between seismic records. Furthermore, modeling predicts that if the
signals were generated exclusively by a single process, such as turbulence or bedload transport, the
spectral differences between OBS2 and OBS3 would exhibit a monotonous decrease in seismic power
with increasing frequency due to signal attenuation (Fig. S1). However, the observed spectral
differences between OBS2 and OBS3 show a non-monotonic behavior. – (line 144).

It is unclear why you are referring to Figure 3c here.

Thank you for pointing this out. The reference to Figure 3c was misplaced and has now been corrected
in the revised manuscript.

(D,E)

Done.

passes -> passing (?)

Done.

378 This allows us to

Done

401 You don't need phrases like "it is important to note that". These make the text harder to read as the
information becomes more dispersed.

We removed "it is important to note that" from the text.

415. It is not clear what log m means for the units. Presumably $\log_{10}(\sigma)$ - please also verify the
base of the logarithm.

We now state in line 424 that we use a log-raised cosine grain size distribution (Tsai et al., 2012), with
a mean grain diameter of 0.02 m and a standard deviation of 0.5.

Supplementary file:

46 absolute -> magnitude

Done.

99 Deploying the OBSs on the terraces outside the thalweg presumably ensures the Scholte wave
propagation speed is fairly uniform spatially and there is limited effect of refraction? Elsewhere, there
might be variations in thickness of the surface sediment layers that might lead to spatial varied wave
speed?

The reviewer correctly notes that variations in sediment properties may cause spatial changes in Scholte
wave propagation speed, potentially leading to wave refraction, and that deploying the OBSs on terraces
outside the thalweg further minimizes the potential for significant lateral velocity variability. However,
even if such velocity variations exist, their impact on arrival times would be minimal—likely limited to
fractions of a second to a few seconds over the source-to-station distances analyzed here. Given that
turbidity current signals span several hours, these minor differences would have no significant effect on
our analysis or methodology.

Reviewer #3

Review of "Ocean-Bottom Seismometers Reveal Surge Dynamics in Earth's Longest-Runout Sediment

Flows” by Kunath et al., submitted to Nature Earth and Environment
In the present paper the authors provide unprecedented observations of sea-floor turbidity currents from
seismic monitoring. The report particularly insightful observational features. Mainly, they are able (i)
to dissociate the turbulence versus sediment transport seismic signatures based on evaluating seismic
power variations with distance, (ii) to identify that the turbidity current operates in 5-30 min long pulses
propagating at varying speeds, and that (iii) the most energetic part of the turbidity current is not
concentrated in the front but rather farther up the flow. These results have implications on the physics
controlling the formation and propagation of these events, which are discussed in the paper. Overall, the
seismic data analysis is thorough, very sound and particularly convincing. The paper is well written and
addresses a particularly interesting and yet poorly known Earth mechanism. For all these reasons I am
strongly in favor of publication in Nature Earth and Environment.

**We thank the reviewer for their thoughtful and positive evaluation of our work.**

I have a few general comments that I think the authors could use to improve the draft, in particular to
make it better suited for Nature Earth and Environment, as well as more specific comments that I provide
below.

GENERAL COMMENTS

The abstract and introduction are good, but I think there is a lack of hierarchy in introducing aspects
related to the key findings of the authors, and how they have the potential to change our view on how
turbidity current operates. At several instances in the abstract and the introduction it is hard to dissociate
what was known before from what the present paper brings in. For example, the typical runout and
velocities seem to be already known, such that the ability of seismic observations to constrain these
aspects might not be the best-selling point to emphasize, or at least not without more details about the
advantages offered by seismic observations (e.g. spatial and temporal scales). Instead, the surge behavior
consisting in repetitive pulses seems to be a new finding with important implications, which appears to
me as comparatively under sold, i.e. under introduced in the abstract and intro, and not sufficiently
discussed in light of existing literature towards the end of the paper.

In addition, I think there is lacking information regarding our knowledge on the seismic signature of
mass movements on earth (rivers, debris flows) and to which extent one might think to extend this to
turbidity currents. There should be at least a short paragraph on this in the introduction and some
extended discussion later on.

**We appreciate the reviewer's suggestion regarding the need to clearly distinguish what is already known
from what our paper contributes. In response we have revised abstract as follows:**

*Turbidity currents carve Earth's deepest canyons, form Earth's largest sediment deposits, and break*
*seabed telecommunications cables. Directly measuring turbidity currents is notoriously challenging due*
*to their destructive impact on instruments within their path. This is especially the case for canyon-*
*flushing flows that can travel >1,000 km at >5 m/s, whose dynamics are poorly understood. We*

deployed ocean-bottom seismometers safely outside turbidity currents, and use emitted seismic signals
to remotely monitor canyon-flushing events. By analyzing seismic power variations with distance and
signal polarization, we distinguish signals generated by turbulence and sediment transport, and
document the evolving internal speed and structure of flows. Flow-fronts have dense near-bed layers
comprising multiple surges with 5-to-30-minute durations, continuing for many hours. Fastest surges
occur 30–60 minutes behind the flow-front, providing momentum that sustains flow-fronts for >1,000
196 km. Our results highlight surging within dense near-bed layers as a key driver of turbidity currents'
long-distance runout. – (line 12-23)

We have also revised the introduction to clearly distinguish between what was previously known and
the contributions of our study.

In line 39: Previous measurements of such flows have come from cable breaks or destroyed moorings
at just two sites (5,17), providing estimates of flow front (transit) speeds and run-out distances, but
offering limited insight into their internal structure.

We have added a new paragraph to explain how seismology has been used to study terrestrial mass
movement processes, describe its current applications in the marine realm, and clarify how our study
extends beyond these previous initiatives (line 49): On land, remote seismic monitoring has
revolutionized our understanding of major geohazards such as floods, debris flows, glacial lake
outbursts, and avalanches, by detecting their ground motions via seismometers with millisecond
precision across distances ranging from hundreds of meters to hundreds of kilometers (27-29). These
data have yielded key insights into how ground motion signals are generated at the source, transmitted
through the environment, and ultimately recorded at seismic stations, thereby advancing process
understanding, disaster response, and early warning systems (29). In submarine settings, ocean-bottom
seismometers (OBSs) and hydrophones have occasionally recorded seismic and acoustic signals from
submarine mass movements (30-33), but their use remains nascent and often limited to detecting
occurrence and overall duration.

Here, we present the first detailed measurements of the internal structure, speed and spatiotemporal
evolution of dense-frontal cells in canyon-flushing turbidity currents, thereby going beyond previous
measurements restricted to just their front-speed, runout-distance or total duration (5,33). These results
were obtained by analyzing seismic signals recorded by OBSs positioned safely outside the Congo
Canyon and Channel off West Africa.

We moved the Congo Canyon and Channel description from the Introduction to a dedicated “Turbidity
currents in the Congo Canyon and Channel” section (line 68). This streamlines the Introduction,
allowing it to remain concise and focused on the key aspects of our findings, and to more clearly connect
the study’s objectives with the knowledge gaps highlighted earlier.

Finally, we have added to the paragraph in line 203 to incorporate a comparison with seismic signals
from terrestrial natural systems. “The overall seismic waveform—an emergent arrival, a maximum, and
a long decay—is not due to the flow front's approach, peak proximity to the station, and subsequent
distancing. The flow front passes the OBS6 30–60 minutes before the peak seismic energy, which rather
corresponds to the fastest surges. These faster-moving surges likely carry higher sediment
concentrations, contributing to their increased speeds. This pattern aligns with observations of debris
flows, where coarse surge fronts generate stronger seismic amplitudes than the later, slower parts of
the flows (39). Consequently, this also explains why faster surges produce stronger seismic signals than
slower surges, even when these surges come from the same back azimuth and thus position along the
canyon-axis. This observation indicates that the highest sediment concentrations and fastest surges
within canyon-flushing turbidity currents are located a significant distance (e.g. 19 km) behind the flow
front.”

I find the discussion line 253-261 on this aspect to be difficult to follow, although this point seems quite
important for the broader implications of the current findings.

We have revised the paragraph to make it easier to follow. We clarify how internal surges within the
turbidity current can deliver momentum to the flow front, potentially altering its speed beyond what a
local force balance (as used in traditional models) would suggest (line 269): Traditionally, turbidity
current front speeds have been modeled based on a local balance between gravitational driving forces
(related to flow thickness, excess density, sediment concentration, and seabed gradient) and frictional
forces at the flow front (10). However, our findings suggest that this perspective omits a critical factor:
momentum transfer from within the flow itself. Specifically, that faster-moving, higher sediment
concentration 'internal surges' can deliver additional momentum to the flow front. This mechanism
implies that front speeds may be influenced not only by local force balances but also by the internal
structure and dynamics of the turbidity current.

SPECIFIC COMMENTS

Line 63: Reference to an EGU presentation to argue something fundamentally important is I think not
appropriate. The authors claim that “the seismic signals originate from the faster-moving frontal zone”
as an already demonstrated finding, but the citation has not been peer-reviewed, thus I don't think it is
appropriate.

Thanks for pointing this out. We have updated this reference to reflect the peer-reviewed publication,
which is available in Geophysical Research Letters.

Line 77 to 81: The list of study objectives is quite complete, but it is hard to connect them with the
various lack of knowledge discussed earlier. I would suggest to make a better connection, for example
by disseminating the objectives after each lacking knowledge paragraphs.

Thank you for your suggestion. We agree that aligning the study objectives with the knowledge gaps is
essential for clarity. However, the Nature guidelines require that the objectives be stated collectively at
the end of the introduction. To address this, we have revised the introduction to make it more concise
and to better highlight the connection between the knowledge gaps and our objectives. Please see our
previous comments (reviewer response file, lines 197 to 222) for further details.

Figure 3: really interesting analysis. Would be nice to have the observed spectrum of OBS3-OBS2, to
be more easily compared with the theoretical one. I assume the authors did not show it because maybe
it did not readily work, but that is ok, the conceptual model and the fact that this feature exists in the
data is convincing, it would just be easier for the reader to evaluate the degree up to which it is indeed
convincing.

The theoretical models for turbulent flow and bedload transport incorporate simplifications when
applied to submarine environments, and key parameters governing sediment transport in turbidity
currents remain poorly constrained, as discussed in line 431. For this reason, we opted not to invert for
these parameters in pursuit of a precise spectral fit, as doing so could undermine the geological validity
of the results. The primary objective of this exercise was not to achieve an exact spectral match or derive
precise quantitative properties of the turbidity current flow, but rather to demonstrate that the observed
spectral characteristics are consistent with two distinct seismic sources operating within separate yet
overlapping frequency bands.

Line 375: The models of Gimbert al and Tsai et al. have been developed for surface waves with no water
layer on top. In the present case the water layer has to be accounted for, which I think is done by the
author, based on the reference to Scholte wave in Table 1. However, I think much more information is
required on how the authors describe this Scholte wave, and how seismic wave amplitude may differ in
this case compared to the classical surface wave case investigated previously.

My comment above regarding lacking information on the Green function actually also applies to the
seismic source description of turbidity current. In particular, the crucial parameter in the theory of
Gimbert et al. is the water flow velocity. The authors must be using a description of water flow velocity
as a function of flow height and surface slope, but they are not explaining this. There is also a lot of
complementary information missing about the application of the model to their present setup. I think
this information must be provided.

The reviewer is correct in noting that the models of Gimbert et al. (2014) and Tsai et al. (2012) assume
Rayleigh waves in the absence of a water layer. While we have not modified these models to explicitly
account for the water layer and incorporate Scholte waves, we believe this has minimal impact on our
qualitative interpretation. Sensitivity tests using numerical modeling indicate that, given our geometry
(~1.5 km source-to-station distance with a water layer present), the seismic wavefield is only slightly

affected compared to the Rayleigh wave scenario (see Fig. 2 for details). Scholte waves, in this case,
 would exhibit amplitudes approximately 20% larger and show minor differences in periods, while their
 overall waveform characteristics remain comparable. Since our primary focus is to demonstrate that two
 distinct seismic sources can explain the observed spectra, rather than achieving precise amplitude
 matches or quantitatively inverting flow parameters, we consider this approach appropriate for the scope
 of our study.

 **Figure 1:** We modeled Rayleigh and Scholte waveforms generated by a vertical, downward force (e.g.,
 particle impact) on the seafloor using a layered velocity structure. The Scholte wave simulation included
 a 2 km-thick water layer, whereas the Rayleigh wave simulation excluded it. Both setups assumed a
 source-to-station distance of 1.5 km. The results demonstrate that the overall waveform characteristics
 are similar, with the Scholte wave exhibiting approximately 20% larger amplitudes and the Rayleigh
 wave package displaying a slightly shorter period.

We updated the sections on the models by Gimbert et al. and Tsai et al., clarifying that these models
 assume Rayleigh waves, whereas the marine environment generates Scholte waves (line 431):
 Understanding the seismic signals generated by turbidity currents requires reconciling key differences
 between theoretical models and the complexities of real-world flows. For instance, while the turbulent
 flow36 and bedload transport37 models assume Rayleigh waves without a water layer, Scholte waves
 are present in our marine setting. This discrepancy could introduce slight differences in waveform
 characteristics, such as seismic amplitude. However, these differences are not expected to
 fundamentally alter the qualitative comparison of spectral shapes and trends.

We have also expanded the description of the turbulence and bed-load transport models, clarifying how
 velocities are defined within these frameworks. Below is a concise, more accessible summary, while the
 full technical details are provided in the original reference, which we also include (line 408): The
 turbulent flow model (36) attributes seismic noise to fluctuating forces from turbulent water flow.
 Pressure and shear stresses exerted on the riverbed by turbulence cause vibrations that propagate as
 Rayleigh waves, detectable by nearby seismic stations. The magnitude and characteristics of these
 forces depend on key hydraulic parameters, including water depth, bed roughness, and flow velocity—
 where flow velocity itself is influenced by the river's slope, channel roughness, and flow depth. The

model treats these forces as stochastic processes to reflect the inherent randomness of turbulence. By
integrating the contributions of these forces over a range of frequencies, the model calculates the Power
Spectral Density (PSD) of the resulting seismic noise, establishing a quantitative link between river flow
properties and their seismic signatures.

Line 152 : reference to this statement ? I have a hard time understanding what these sentences relate to,
previous work or current one, if the later then refer to figure?

We are now referring to Fig. 4.

Line 224: remove “into”

Done.

L262: You might consider citing Cook et al., 2021, who observed a river flood also with seismics
constituted by a flow front that’s less energetic than a later arriving flow component, which the authors
attributed to a more concentrated sediment wave. In that case, however, the second arriving flow is
slower than the front, which is different than observed here, but maybe tells us something about the
similitudes/differences in the underlying physics?

We thank the reviewer for suggesting Cook et al. (2021), which demonstrates how terrestrial seismology
can advance our understanding of GLOF dynamics. In their study, the fast-moving water bore precedes
the slower, sediment-rich wave, illustrating two distinct phases with different flow characteristics.
However, this externally triggered, multi-pulse behavior contrasts fundamentally with the internally
driven surges of turbidity currents.

Our focus at L262 (now 286) is on the lag time or distance of peak velocity behind the flow front in
canyon-flushing turbidity currents, a phenomenon not previously quantified via observation. Based on
our data, we found differences between observations and assumptions in existing models. Since Cook
et al. (2021) does not directly inform this specific point, we have included the reference in the
introduction to showcase how seismology has advanced the understanding of terrestrial geomorphic
events, but have opted not to include it in the discussion at L286.